# Cancer-Associated Fibroblasts Influence Survival in Pleural Mesothelioma: Digital Gene Expression Analysis and Supervised Machine Learning Model

**DOI:** 10.3390/ijms241512426

**Published:** 2023-08-04

**Authors:** Sabrina Borchert, Alexander Mathilakathu, Alina Nath, Michael Wessolly, Elena Mairinger, Daniel Kreidt, Julia Steinborn, Robert F. H. Walter, Daniel C. Christoph, Jens Kollmeier, Jeremias Wohlschlaeger, Thomas Mairinger, Luka Brcic, Fabian D. Mairinger

**Affiliations:** 1Institute of Pathology, University Hospital Essen, University of Duisburg Essen, 45147 Essen, Germany; sabrina.borchert@uk-essen.de (S.B.); alexander.mathew98@yahoo.de (A.M.); alina.nath@stud.uni-du.de (A.N.); michael.wessolly@uk-essen.de (M.W.); elena.mairinger@gmail.com (E.M.); daniel.kreidt@stud.uni-due.de (D.K.); robert.walter@uk-essen.de (R.F.H.W.); 2Center for Pathology in Essen-Mitte, 45131 Essen, Germany; j.steinborn@pathologie-essen.de; 3Department of Medical Oncology, Evangelische Kliniken Essen-Mitte, 45131 Essen, Germany; d.christoph@kliniken-essen-mitte.de; 4Department of Pneumology, Helios Klinikum Emil von Behring, 14165 Berlin, Germany; jens.kollmeier@helios-gesundheit.de; 5Department of Pathology, Diakonissenkrankenhaus Flensburg, 24939 Flensburg, Germany; wohlschlaegerje@diako.de; 6Department of Tissue Diagnostics, Helios Klinikum Emil von Behring, 14165 Berlin, Germany; thomas.mairinger@helios-gesundheit.de; 7Diagnostic and Research Institute of Pathology, Medical University of Graz, 8036 Graz, Austria; luka.brcic@medunigraz.at

**Keywords:** pleural mesothelioma, digital gene expression, cancer-associated fibroblasts, survival, machine learning

## Abstract

The exact mechanism of desmoplastic stromal reaction (DSR) formation is still unclear. The interaction between cancer cells and cancer-associated fibroblasts (CAFs) has an important role in tumor progression, while stromal changes are a poor prognostic factor in pleural mesothelioma (PM). We aimed to assess the impact of CAFs paracrine signaling within the tumor microenvironment and the DSR presence on survival, in a cohort of 77 PM patients. DSR formation was evaluated morphologically and by immunohistochemistry for Fibroblast activation protein alpha (FAP). Digital gene expression was analyzed using a custom-designed CodeSet (NanoString). Decision-tree-based analysis using the “conditional inference tree” (CIT) machine learning algorithm was performed on the obtained results. A significant association between FAP gene expression levels and the appearance of DSR was found (*p* = 0.025). DSR-high samples demonstrated a statistically significant prolonged median survival time. The elevated expression of *MYT1*, *KDR*, *PIK3R1*, *PIK3R4*, and SOS1 was associated with shortened OS, whereas the upregulation of *VEGFC*, *FAP*, and *CDK4* was associated with prolonged OS. CIT revealed a three-tier system based on *FAP*, *NF1*, and *RPTOR* expressions. We could outline the prognostic value of CAFs-induced PI3K signaling pathway activation together with FAP-dependent CDK4 mediated cell cycle progression in PM, where prognostic and predictive biomarkers are urgently needed to introduce new therapeutic strategies.

## 1. Introduction

Desmoplastic stromal reaction (DSR), containing cancer-associated fibroblasts (CAFs), is a well-known phenomenon in various carcinomas [1,2]. It is defined as a newly formed tumor-associated fibroblastic stroma surrounding invasive tumor cells [3]. Normal epithelial cells are segregated from fibroblasts and dispersed as single cells within the stromal tissue compartment in the adjacent connective tissue [4]. During homeostasis, quiescent fibroblasts show minimal metabolic and transcriptional activity but play a pivotal role in defining the differentiation status of adjacent epithelia, via the secretion of specific signaling factors [4]. The activation of fibroblasts in a non-homoeostatic condition seems to occur through different mechanisms: stress signaling from inflammation and (chronic) wound healing, or through mechano-transduction [4,5,6,7,8]. Acute inflammation during wound healing is a classic example of the separation loss between epithelial and stromal cell populations. It is characterized by the transition of resting fibroblasts into activated ones (myofibroblasts) [4,6]. The release of growth factors, such as TGF-β, EGF, PDGF, and FGF2 by injured endothelial cells and resident macrophages, is assumed to be the main driving force for this transformation [6]. It eventually results in fibroblasts sprouting contractile fibers, while also expressing α-smooth muscle actin (α-SMA) and the ED-A domain of fibronectin 1 (FN1) [5]. Furthermore, physical stimulation of resident fibroblasts, leading to the chemical translation of mechanic signals by releasing the aforementioned factors, is commonly accepted as a possible explanation for fibroblast activation [4,5,6]. Despite indications in multiple studies that CAFs produce fibrotic stroma, the exact mechanism of its formation has not been adequately clarified. DSR is present in numerous tumors and also has a prognostic significance owing to its association with poorer prognoses [3,9,10].

A local immune response in the pleura to undecomposed, inhaled microfibers, released from corroding asbestos, has been identified as a main driving force of tumorigenesis of the pleural mesothelioma (PM) [11]. Thereby, both the persistent asbestos fibers and continuous inflammatory reaction in the tissue provide mutagenic surroundings. Evasion of apoptosis and the accumulation of pro-oncogenic mutations in the affected mesothelial cells induces malignant transformation and the development of PM [12,13]. By using persistent modulation of the local inflammatory reaction to their benefit, e.g., by releasing proinflammatory cytokines such as TGF-β, these malignant cells can awaken the resting fibrocytes, which switch their phenotype into activated myofibroblasts, defined as CAFs [14,15]. In addition to the paracrine signaling between cancer cells and CAFs, which has already been shown to play an integral part in tumor progression [16], DSRs, otherwise known as stromal changes, were themselves identified as factors for a poor prognosis in PM patients [17].

Similar to CAF-mediated changes in fiber orientation, the density and structure are known to impede the effectiveness of chemotherapy in some tumors [18,19,20,21]; therefore, we aimed to assess the impact of CAF and DSR presence on survival in PM.

## 2. Results

### 2.1. Histologic and Immunohistochemical Evaluation

Altogether, DSR in hematoxylin–eosin (HE) stained slides were observed in 52 of the 77 (67.5%) analyzed patient samples. Furthermore, 35/77 (45.5%) of these were classified as DSR-low and 17/77 (22.1%) as DSR-high. Figure 1A,B depict an example of IHC and HE staining with DSR-low, while Figure 1B,C depict an example of DSR-high.

In the 25 tumors where DSR was absent in the HE staining, only 5 specimens were completely negative for FAP, with the remaining 20 samples presenting only a few positive cells scattered within the tumor area. In the 35 DSR-low samples, 9 were scored as 1, 17 as 2, and 9 as 3. On the other hand, in the DSR-high samples, 4 were scored as 1, 8 as 2, and 5 as 3.

Overall, 5 samples were defined with a score of 0 (7.7%), 21 had a score of 1 (32.3%), 25 had a score of 2 (38.4%), and 14 had a score of 3 (21.5%). The histological observations are summarized in Table 1. In 12 samples (15.6%), at least 1 tumor core did not illustrate representative signals and these were, therefore, excluded.

### 2.2. Association between DSR, FAP Immunohistochemistry, Gene Expression, and Survival

A significant association was found between higher *FAP* gene expression levels and the appearance of DSR (*p* = 0.025). However, there was no significance between FAP immunohistochemistry and DSR (*p* = 0.23). Nevertheless, a strong significant association was found between FAP immunohistochemistry and *FAP* gene expression (*p* = 0.006). Interestingly, two samples with a FAP score of 1 classification showed high counts of *FAP* gene expression.

Neither the appearance of DSR (*p* = 0.873; HR: 1.04), its amount (*p* = 0.608; HR: 1.21), nor the FAP IHC (*p* = 0.806; HR: 1.04) revealed a significant association with OS. However, the median survival time (29.2 months; 95% CI: 17.1–44.2) of DSR-high samples was prolonged compared to those without DSR (15.9 months; 95% CI: 13.8–27.3) or with low DSR (15.8 months; 95% CI: 11.4–24.4), thereby reaching a significant difference (*p* = 0.044; HR: 1.75; 95% CI: 1.02–3.00).

For PFS, no significant association was found for the appearance of DSR (*p* = 0.597), its amount (*p* = 0.873), or the FAP IHC (*p* = 0.807). In contrast to OS, a comparison between DSR-high and the other two groups revealed no significant differences in PFS (*p* = 0.424). These results are summarized in Appendix A.

### 2.3. OS and Digital Gene Expression Analysis

When analyzing gene expression counts as a continuous variable, eight genes were identified as being associated with OS. The elevated expression of *MYT1*, *KDR*, *PIK3R1*, *PIK3R4*, and *SOS1* was associated with a shortened OS, whereas the upregulation of *VEGFC*, *FAP*, and *CDK4* was associated with a prolonged OS. An overview of all calculated *p*-values related to OS and gene expression can be found in Appendix A.

In addition, according to the expression of those eight genes, samples were categorized into either the high- or low-expressing group. Then, the optimal cutoff for the best separation between groups was chosen for each gene.

The *MYT1* high expression group demonstrated a median OS of 13.3 months (*n* = 16; 95% CI: 8.9–27.3) versus 21.6 months (*n* = 57, 95% CI: 16.4–28.2) in the low expression group (cutoff: 10 counts—present vs. absent; *p* = 0.003; HR: 0.42; 95% CI: 0.24–0.76). The two-year OS rate in the high vs. low expression group was 18.8% and 43.9%, respectively. The *KDR* (cutoff: 900 counts; *p* = 0.001; HR: 0.36; 95% CI: 0.19–0.68) analysis revealed a median OS in the high expressing group of 11.4 months (*n* = 13; 95% CI: 6.7–n.a.) versus 18.9 months (*n* = 60; 95% CI: 15.9–28.5) in the low expressing group. The two-year OS rate was 15.4% and 43.3% in the high vs. low expressing group. For *PIK3R1* (cutoff: 1150 counts; *p* = 0.004; HR: 0.27; 95% CI: 0.10–0.69), the median OS in the high expressing group was 12.1 months (*n* = 5; 95% CI: 6.2–n.a.), whereas in the low expressing group, it was 18.7 months (*n* = 68; 95% CI: 15.9–27.3). The two-year OS rate in the low-expressing group was 42.2%, whereas none survived for that long in the high-expressing group. For *PIK3R4* (cutoff: 175 counts; *p* = 0.01; HR: 0.42; 95% CI: 0.21–0.85), the median OS in the high expressing group was 14.5 months (*n* = 10; 95% CI: 14.5–n.a.) versus 18.9 months (*n* = 63; 95% CI: 15.9–28.2) in the low expressing group. It was also reflected in the two-year OS rates, which were 12.7% and 41.3%, in the high and low expressing groups, respectively. *SOS1* analysis (cutoff: 400 counts; *p* = 0.039; HR: 0.59; 95% CI: 0.35–0.98) presented a median OS in the high expressing group of 15.9 months (*n* = 25; 95% CI: 15.2–22.4) versus 20.4 months (*n* = 48; 95% CI: 14.4–30.3) in the low expressing group. Additionally, the two-year OS rate was 24.0% and 45.8% in the high vs. low expressing group, respectively.

The high *VEGFC* expression group (cutoff: 60 counts; *p* = 0.008; HR: 1.93; 95% CI: 1.18–3.18) demonstrated a median OS of 27.3 months (*n* = 29; 95% CI: 15.6–37.0) versus 15.9 months (*n* = 44; 95% CI: 13.3–22.2) in the low expression group. Furthermore, the two-year OS rate was 51.7% and 29.6% in the high and low expression groups, respectively. For the FAP (cutoff: 95 counts; *p* = 0.012; HR: 1.93; 95% CI: 1.15–3.24), a median OS in the high expressing group was 18.7 months (*n* = 46; 95% CI: 14.4–30.9) versus 16.0 months (*n* = 27; 95% CI: 15.2–22.4) in the low expressing group. The two-year OS rate in the group with high expression was 45.6% vs. 25.9% in the low expression group. The *CDK4* high expression group had a median OS of 18.9 months (*n* = 60; 95% CI: 15.6–28.2) versus 13.6 months (*n* = 13; 95% CI: 7.2–n.a.) in the low expression group (cutoff: 105 counts; *p* = 0.008; HR: 2.31; 95% CI: 1.22–4.35). These results are summarized in Appendix A. The two-year OS rates were 43.3% and 15.4%, for the high and low expression groups, respectively.

Kaplan–Meier curves for each defined group are presented in Appendix A. Visualizations of the HR, 95% CI, and calculated *p*-values are depicted as an overview in Figure 2.

### 2.4. PFS and Digital Gene Expression Analysis

Three CAF-associated genes were identified as being associated with PFS, following the analysis of the gene expression counts as a continuous variable. The elevated expression of *PIK3C3* and *CDKN1B* was associated with shortened PFS time, whereas the upregulation of NRAS was associated with prolonged PFS. An overview of all the calculated *p*-values related to PFS and gene expression is presented in Appendix A.

Based on the expression of those three genes, samples were stratified into either a high- or low-expression group. The cutoffs that resulted in the best separation between the groups were chosen for each gene.

For *PIK3C3* (cutoff: 200 counts; *p* < 0.001; HR: 0.24; 95% CI: 0.10–0.59), the median PFS in the high expressing group was 7.4 months (*n* = 16 with 14 events; 95% CI: 4.1–8.6) versus 9.7 months (*n* = 18 with 15 events; 95% CI: 9.2–23.1) in the low expressing group. Moreover, the one-year PFS rate differed between 0% and 38.5% for the high and low expression groups, respectively. *CDKN1B* in the high expression group demonstrated a median PFS of 4.1 months (*n* = 6 with 4 events; 95% CI: 3.9–n.a.) versus 8.9 months (*n* = 28 with 25 events; 95% CI: 8.3–10.8) in the low expression group (cutoff: 600 counts; *p* = 0.02; HR: 0.25; 95% CI: 0.07–0.88). The one-year PFS rate was 0% and 23.3% in the high vs. low expression groups, respectively. For *NRAS* (cutoff: 113 counts; *p* = 0.001; HR: 5.23; 95% CI: 1.74–15.72), the median PFS in the high expressing group was 9.3 months (*n* = 22 with 21 events; 95% CI: 8.5–14.3) versus 6.5 months (*n* = 12 with 8 events; 95% CI: 4.0–n.a.) in the low expressing group. The one-year PFS was 27.3% and 0% in the high and low expression groups, respectively.

The Kaplan–Meier curves for all the defined groups can be found in Appendix A. A visualization of the HR, 95% CI, and calculated *p*-values is depicted as an overview in Figure 2.

### 2.5. Decision-Tree-Based Analysis of OS and PFS

Decision-tree-based analysis using the “conditional inference tree” (CIT) machine learning algorithm for the OS revealed a three-tier system based on the *FAP*, *NF1*, and *RPTOR* expression levels.

Analysis of the *FAP* expression and its best cutoff (calculated cutoff: 97 counts; *p* = 0.005) revealed that the group with the *FAP* overexpression had the best OS. Furthermore, in those samples where *FAP* expression was beneath the cutoff, the expression of *NF1* (calculated cutoff: 362 counts, *p* = 0.038) was used to identify the subgroup with the shortest OS time. On the other hand, in the samples below the *FAP* and above the *NF1* expression cutoffs, the expression of *RPTOR* (calculated cutoff: 303 counts, *p* = 0.044) was used to divide them into two separate subgroups. The calculated tree is presented in Figure 3A.

Regarding PFS, a two-tier scoring system based on the gene expression levels of *TGFBR1* and *MAP2K1* was identified. The group overexpressing *TGFBR1* (calculated cutoff: 484 counts; *p* = 0.022) was identified as the group with the best PFS. In samples where the *TGFBR1* expression was beneath the cutoff, the *MAP2K1* (calculated cutoff: 210 counts; *p* = 0.040) was used to divide the groups into subgroups with the shortest (above cutoff) and medium (beneath cutoff) PFS times. The tree is illustrated in Figure 3B.

## 3. Discussion

The complex interactions in extracellular signaling in the TME and between TME and tumor cells have begun to receive more and more attention. The clinical impact of the biological mechanisms of DSR, such as the production of fibrotic stroma by CAFs, has not yet been adequately clarified in PM. Using histopathologic analyses, digital gene expression analysis, and supervised machine learning, we have demonstrated that CAFs and their related genes are important for predicting OS and PFS.

Growing knowledge of cancer biology shows that the therapeutic response, resistance, and outcome depend not only on the histology, single-gene deregulation, or single hallmark of cancer. Multigene expression analyses are a promising strategy for the given complexities. In this context, upfront diagnostics of biomarker expressions will be required, and certain histology-independent phenotypes might become drivers of clinical decision-making. Thus, highly sophisticated bioinformatical and statistical methods are needed for solid data curation and biomarker identification.

In previous research, we presented machine-learning techniques that improved the prediction of responses to immunotherapy, thereby enabling rapid and precise clinical decision-making [22,23]. Similar techniques have been presented previously by other groups, using machine learning to identify *IFNγ*-related mRNA profiles [24] and the 18-gene tumor inflammation signature [25] to predict responses to immunotherapy. Our present study used CIT-based survival classifiers to model the decision-tree dependent two- and three-tier scoring systems.

Our decision-tree analysis revealed *TGFBR1* and *MAP2K1* as key markers for progression-free survival in the investigated PM patients. TGF-β has been described to modulate PI3K/AKT signaling [26]. The phosphoinositide 3-kinases (PI3Ks) are a large family of lipid enzymes that are able to phosphorylate the 3′-OH group in the phosphatidylinositol on the plasma membrane [27]. These enzymes are activated downstream of the tyrosine kinase receptors and/or G protein-coupled receptors. Through AKT, they can induce the stimulation of the mammalian target of rapamycin (mTOR) [27]. PI3Ks play a key role in signaling different cellular processes, such as normal metabolism, inflammation, cell survival, motility, and cancer progression [28]. Mutations often induce a gain of function or hyperactivity of PI3Ks in tumors [27]. Published data also indicate that cytokine-mediated activation of PI3K–AKT signaling inhibits cisplatin-induced cell cycle arrest [29]. The PIK3R1 gene, which encodes for p85α, is the regulatory subunit of class I PI3K [30,31]. PIK3R1 mutations have been reported in several types of cancer, including endometrioid endometrial cancers, non-endometrioid endometrial cancers, glioblastomas, breast, ovarian, and colon tumors [32,33]. The mutations primarily accumulate in the inter-SH2 (iSH2) domain and involve residues that interact with the C2 domain of the catalytic subunit p110α, which is encoded by the PIK3R4 gene [31,34]. PTEN encodes for the phosphatidylinositol-3,4,5-trisphosphate 3-phosphatase [35] and primarily dephosphorylates phosphoinositide substrates. It negatively regulates phosphatidylinositol-3,4,5-trisphosphate intracellular levels and functions as a tumor suppressor by negatively regulating the AKT/PKB signaling pathway [36]. In our study, we found that the overexpression of *PIK3R1* and *PIK3R4* was associated with a reduction in OS. We assume that the promotion of AKT phosphorylation results in the destabilization of the PTEN protein, which promotes tumor progression.

Furthermore, we have also observed an association between *PIK3C3* overexpression and shortened PFS. The *PIK3C3* gene encodes for the only known class III PI3-kinase member, named vacuolar protein sorting 34 (Vps34) [27]. Vps34 acts positively on mTOR/S6K1, integrating glucose and amino acid inputs into the mTOR pathway. Activation of the mTOR pathway promotes the proliferation of tumor cells [37]. Our observation concurs with the mechanisms previously described in the literature. It also underlines the importance of understanding the functions and interactions of the investigated genes. Notably, recent studies reported that targeting the PI3K/mTOR pathway could act as a potentially effective therapeutic strategy in mesothelioma [38].

Within an analysis of factors significantly affecting the OS in our patient cohort, we have also analyzed the tumor microenvironment, more precisely, the impact of CAFs. We found that a higher degree of desmoplasia was associated with higher FAP IHC expression and interpreted desmoplastic changes as a result of stromal changes induced by CAFs. Interestingly, *FAP* gene expression correlated to the strength/amount of DSR, whereas FAP IHC did not. Moreover, both high levels of DSR and *FAP* gene expression showed a significant correlation with patient survival, whereas this association was not present for FAP IHC. In our opinion, the reason for this discrepancy lies in the FAP IHC scoring system, when evaluating the percentage of FAP-positive stromal cells. In contrast to digital gene expression analysis, which results in an overall count of *FAP* mRNA molecules, the IHC scoring system does not take into account the actual strength of the FAP protein expression in positive cells. In this way, the exact phenotype of the analyzed cells is not adequately presented. Additionally, we have to consider the chronological and spatial aspects of the CAFs lifecycle within the evolving tumor stroma. They present with a quiescent phenotype in homeostatic fibroblasts, a highly proliferative phenotype in the proliferating phase, and a senescent but secretory one in the final stage. The phenotype of CAFs changes in reversed order from the tumor center to the invasive tumor front [4]. This is represented by different expressions of activation markers such as *FAP*, *FN1*, and *ACTA*. Therefore, this difference in differentiation and the overall DSR of the tumor is much better reflected by an absolute FAP quantification of protein or gene expression level than by the number of FAP-positive fibroblasts, independent of their staining intensity.

Interestingly, increased FAP expression, and thus, CAF presence resulted in a significant survival advantage within our cohort. We think that this might result from another factor correlating with higher desmoplasia, and in our cohort with a better OS and *CDK4* expression. Cisplatin, the current backbone of most PM therapies, targets fast-proliferating cells during the transfer from G1-phase into S-phase in mitosis [39,40]. Since the activity of CDK4 is restricted to the G1–S phase transfer [41], an increased level of this kinase might increase the efficacy of the treatment, thereby positively influencing survival rates. This is a rather controversial finding since the current literature predominantly assigns CAFs to the role of mediating chemotherapy resistance in PM [42]. Moreover, another factor associated with desmoplasia and in our study with an improvement in OS was *VEGFC*, which is also considered a negative prognostic marker [43]. Since our data indicate a completely different prognostic value, further research in this field is needed to fully comprehend the value of the molecular alterations in PM. Nevertheless, this might be a valid explanation as to why clinical trials, including VEGF inhibition in previously untreated PM patients with high VEGF expression, fail to meet expectations [44].

Our study has several limitations. Firstly, the limited number of analyzed genes in the used custom panel does not cover the whole span of the complex inter-relations in paracrine signaling in the different parts of the tumor microenvironment. Furthermore, the hypothesis-driven selection of target genes may be more error-prone in comparison to the supervised approach. Therefore, some of our conclusions remain putative, although the prognostic impact of the single markers revealed in our analyses remains factual. On the other hand, we have only analyzed gene expressions and not the actual protein/enzyme quantities. For the cutoff calculation for the gene expression results, we used unifactorial CIT with the respective bucket in the terminal node size fitting as our aspired statistical power. Nevertheless, in this case, we decided to target biologically relevant groups, although the sample size is quite small. Therefore, we started with a density analysis of the count distribution for each target across all samples and tried to determine the distribution curves separating the different biological meaningful classes (low to absent, middle, high expression, etc.). We tried to determine cutoffs for the most likely biologically meaningful groups in the crossing points between the different distribution curves. These specific cutoffs may not lead to the strongest possible statistical results; however, we think they are much more adaptable to other gene expression datasets, as the numerically specific cutoffs to achieve the best differentiation in this cohort may not be reproducible in those cases. Nevertheless, we are convinced that this separation of potential biological meaningfulness can be used as a better standard, which can be compared in other studies.

Moreover, the use of bulk RNA analysis is challenging because it does not allow the assignment of cell types. However, we think that it is the right approach for spatial transcriptomics, where small-scale applications are challenging to implement into routine diagnostics. All the data in the translational approach should be adaptable for use in common and widely applied methods, such as targeted RNA bulk analysis.

To better understand the given relations, functional validation of the results, e.g., in vitro experiments that manipulate the identified targets remains necessary to assess their clinical benefit.

## 4. Materials and Methods

### 4.1. Patient Cohort

This retrospective study is based on formalin-fixed paraffin-embedded (FFPE) tumor specimens from 77 patients diagnosed with PM. Tumors were re-classified according to the 2015 WHO classification [45] and staged according to the 2017 TNM classification [46]. Initial diagnosis and the appearance of DSR were evaluated visually by two experienced pathologists (JW, JS). If >10% and <50% of the tumor area was covered by stromal cells, it was classified as DSR-low. Samples with at least 50% of the tumor area covered by stromal cells were classified as DSR-high. All patients were treated at the West German Cancer Centre or the West German Lung Centre (Essen, Germany) between 2006 and 2009 or at the Helios Klinikum Emil von Behring (Berlin, Germany) between 2002 and 2009. Overall survival (OS) was available for 73 patients, all deceased at the time of data collection, while progression-free survival (PFS) was available for 34 patients showing 29 events. PFS was defined from the start of the treatment until the first radiologically determined progression (modRECIST). Overall survival was defined from the initial diagnosis until death or loss of follow-up. This information has been added to the description of the patient cohort in the Materials and Methods section. Therapeutic response was evaluated using the modified Response Evaluation Criteria in Solid Tumors (modRECIST) to assess radiological response in PM [47,48]. It was assessed for 70 tumors with 40 documented progressions.

The study included 62 patients with epithelioid, 8 with biphasic, and 7 with sarcomatoid PM. All specimens were collected before systemic treatment. Patients’ clinicopathological data are summarized in Table 2.

### 4.2. Immunohistochemistry

Immunohistochemistry (IHC) was performed on tissue microarrays (TMA) containing 3 cores, with a diameter of 0.6 mm from different areas of each tumor specimen to overcome tumor heterogeneity, and, when feasible, 1 core containing normal lung tissue and unaffected pleura. The immunohistochemical staining for the fibroblast activating protein (FAP) with primary monoclonal antibody (clone SP325, Abcam, Cambridge, UK, dilution 1:100) was performed using an automated staining system (Ventana Discovery XT, Munich, Germany), according to the standard protocol. Briefly, pretreatment for antigen retrieval was performed by heating in citrate buffer (Ultra Cell Conditioning Solution II, Ventana Medical Systems, Basel, Switzerland) at pH 6 and 90 °C for 48 min followed by antibody incubation for 60 min at 36 °C.

After validating on reference tissues (myofibroblasts in the normal dermis) and on normal pleura samples as the negative control, every case was assessed semi-quantitatively. The percentage of FAP-positive stromal cells was evaluated according to Henry et al. [49] and classified in the following way: negative (score 0) = 0%; low (score 1) = 1–10%; moderate (score 2) = 11–50%; high (score 3) = >50%.

### 4.3. RNA Isolation and Quantification

RNA was isolated from 10 µm thick FFPE whole slide sections using the Maxwell RSC RNA FFPE Kit (Promega GmbH, Mannheim, Germany). RNA was eluted in 50 µL of RNase-free water and then stored at −80 °C. RNA was immediately quantified before gene expression analysis using a Qubit Fluorometer (Thermo Fisher Science, Waltham, MA, USA), according to the manufacturer’s instructions, by the appertaining RNA broad range assay kit.

### 4.4. Digital Gene Expression Analysis

To evaluate differences in RNA expression patterns, a custom-designed CodeSet comprising 76 key players in the cell cycle, PI3K-, MAPK-, Wnt-, TGF-β-signaling pathways, growth factors, and markers for CAF were used. The CodeSet contains five reference genes (ACTB, B2M, GAPDH, RPL19, and RPLP0) previously identified as stably expressed in PM. All targets are listed in Appendix A.

The CodeSets and reagents were designed and synthesized by NanoString Technologies (Seattle, WA, USA). Ultimately, 200 ng of each sample was processed. Post-hybridization processing was performed using the nCounter MAX/FLEX System (NanoString), and cartridges were scanned by the Digital Analyzer (NanoString). Samples were preprocessed by the NanoString nCounter PrepStation, using the high-sensitivity program, and cartridges were read at maximum sensitivity (555 FOV).

### 4.5. NanoString Data Processing and Statistical Analysis

NanoString data processing and statistical analysis were performed using the R statistical programming environment (v4.0.2). NanoString data were processed by the NanoStringNorm (version 1.2.1.1) and NAPPA package (version 2.0.1).

Factorial normalization as well as biological normalization were performed with the geometric mean for all reference genes, as described in a previous study [50].

For statistical analysis, several tests were carried out prior to exploratory data analysis: Shapiro–Wilks test; Wilcoxon Mann–Whitney rank sum test or a two-sided Student’s *t*-test; Kruskal–Wallis test or ANOVA; Fisher’s exact test; Pearson’s Chi-squared test and Spearman’s rank correlation and Pearson’s product-moment correlation were performed, as described elsewhere [50].

OS and PFS were assessed by generating single-factorial and combined Kaplan–Meier curves. Cox regression (COXPH-model) was used for survival analysis, and the likelihood ratio test, Wald test, and Score (log-rank) test were used to test for statistical significance. Kaplan–Meier curves and forest plots with a confidence interval (CI) of 95% were considered based on present survival data, and combined survival curves were implemented.

The statistical significance of the results was defined as *p* ≤ 0.05 after adjustment.

### 4.6. Machine Learning

Adaption of profiles was modeled by the supervised machine learning tool conditional interference trees (CTree), as implemented in the “party” library of R [51], using leave-one-out cross-validation. The CTree is a non-parametric class of regression trees leading to a non-parametric class of tree-structured regression models that embed a conditional inference procedure, applicable to all kinds of regression problems, including nominal, ordinal, numeric, and censored as well as multivariate response variables and arbitrary measurement scales of the covariate [51].

## 5. Conclusions

We outlined the prognostic value of CAFs-induced by PI3K signaling pathway activation together with FAP-dependent CDK4-mediated cell cycle progression in PM. Analyzing the role of paracrine signaling within the tumor microenvironment is important to understand the underlying mechanisms involved in tumor progression and the biology of this severe disease. The identification of new prognostic and predictive biomarkers is urgently needed to introduce new therapeutic strategies and improve the clinical management of patients.

## Figures and Tables

**Figure 1 ijms-24-12426-f001:**
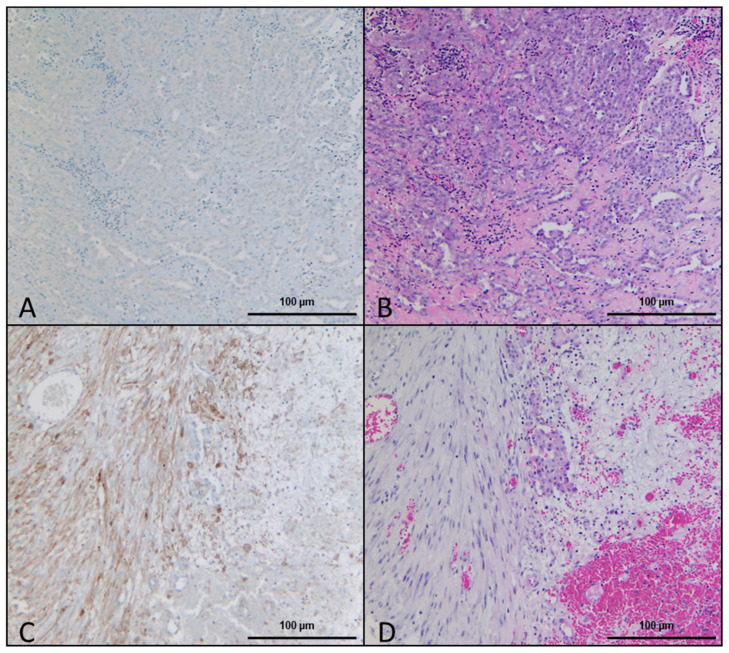
DSR-high and DSR-low. (**A**) Immunohistochemical staining of FAP showing DSR-low and the respective HE staining of the sample in (**B**). (**C**) Immunohistochemical staining of FAP in a sample with DSR-high and the respective HE staining in (**D**).

**Figure 2 ijms-24-12426-f002:**
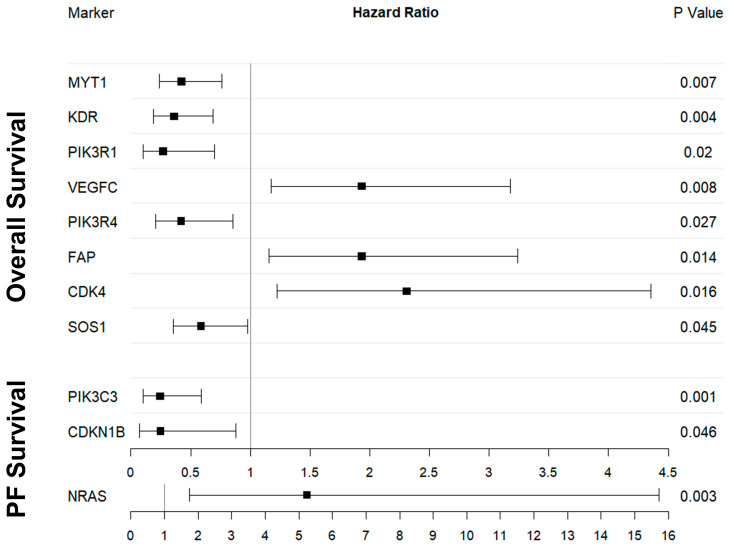
Calculated *p*-values and hazard ratio presentation for the survival analysis correlation with analyzed genes. One can appreciate the positive correlation of *VEGFC*, *FAP*, and *CDK4* with prolonged survival.

**Figure 3 ijms-24-12426-f003:**
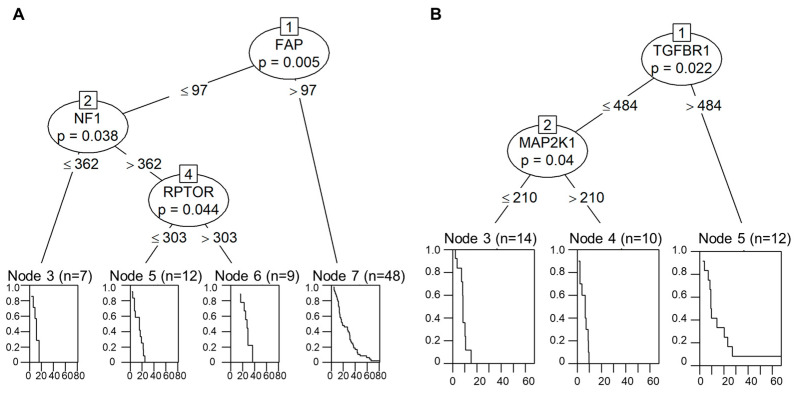
Presentation of decision-tree-based analysis using the “conditional inference tree” machine learning algorithm. For overall survival, a three-tier system based on *FAP*, *NF1*, and *RPTOR* was identified (**A**). For progression-free survival, a two-tier scoring system based on *TGFBR1* and *MAP2K1* was identified (**B**).

**Table 1 ijms-24-12426-t001:** Summary of the histological observation in samples with FAP staining.

Histological Observation.	Percentage of Samples	Number of Samples
Overall DSR in HE-stained slides	67.5%	52/77
DSR-low	45.5%	35/77
DSR-high	22.1%	17/77
DSR absent in HE staining	32.5%	25/77
FAP-negative samples in DSR absent samples	20%	5/25
Score of 1 in DSR-low samples	25.7%	9/35
Score of 2 in DSR-low samples	48.6%	17/35
Score of 3 in DSR-low samples	25.7%	9/35
Score of 1 in DSR-high samples	23.5%	4/17
Score of 2 in DSR-high samples	47.1%	8/17
Score of 3 in DSR-high samples	29.4%	5/17
Score of 0 in overall samples	7.7%	5/65
Score of 1 in overall samples	32.3%	21/65
Score of 2 in overall samples	38.4%	25/65
Score of 3 in overall samples	21.5%	14/65

**Table 2 ijms-24-12426-t002:** Patient’s clinicopathological data.

Number of Patients	77
Gender	
Male	64
Female	13
Histological subtype	
Epithelioid	62
Biphasic	8
Sarcomatoid	7
Age	
Mean/median age at diagnosis (years)	64.6/65.2
Range (years)	37.6–82.9
OS	
Deceased	76
Alive	0
Loss of Follow-Up	1
Median/mean OS (months)	17.1/22.2
95% CI	15.2–24.4
Range (months)	3.1–80.6
PFS	
Partial remission (initial)	3
Stable disease (initial)	32
Progressive disease (initial)	40
Unknown response	2
Median/mean PFS (months)	8.6/10.0
95% CI	7.4–9.7
Range (months)	1.2–67.2

## Data Availability

Data are available upon reasonable request.

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
