# Peer review of "Cancer-Associated Fibroblasts Influence Survival in Pleural Mesothelioma: Digital Gene Expression Analysis and Supervised Machine Learning Model"

_ijms, 2023, doi:10.3390/ijms241512426_

Round 1

Reviewer 1 Report

This manuscript evaluated desmoplastic reaction, FAP protein expression, and several genes in primary mesothelioma. The manuscript has valuable information, but it is hard to read. There are histological and immunohistochemical evaluations, but no figures are shown. The gene expression data with clinical data could be uploaded as a matrix file so the analyses could be confirmed/reproduced.

Specific comments:

1)     In the abstract, could you please add “Fibroblast Activation Protein Alpha” before “FAP”?

2)     Lines 46-65. Is it possible to make a figure summarizing the pathological mechanisms of the production of desmoplastic stromal reaction in cancer?

3)     Lines 66-67. It is stated that DSR associates with prognosis, but, to favorable or unfavorable?

4)     Line 70. Is asbestos the only cause of PM?

5)     Line 321. Does the new 2021 WHO classification of lung tumors provides a necessary update about PM? The manuscript states that the 2015 classification was used.

Please refer to: https://www.jto.org/article/S1556-0864(22)00026-0/fulltext

6)     Line 323. Is it possible to see histological images of DSR low and high?

7)     What are the criteria of OS and PFS?

8)     In Table 1, “unknown gender” = 0 is not necessary.

9)     Regarding Table 1, “lost-of-FU”. Is “FU”, “follow-up”?

10)  Why OS is described by median, mean, and range; but PFS by median, mean, and 95%CI?

11)  Is the diameter of the TMA of 0.6 mm correct? The evaluation area may be very small, even with 3 cores. What about histological heterogeneity within and between samples?

12)  Does the percentage of FAP-positive cells in the stroma refer to the total cells of the sample, including PM? Could you please show pictures?

13)  I am sorry but I cannot find the supplementary Table 1, so I cannot check the list of the 76 genes of the nCounter multiplex Custom CodeSet (NanoString).

14)  Is the data processing of lines 379-389 the recommended procedure of NanoString?

15)  Lines 388-405. Could you please add the links for the different R applications that were used for the biostatistics? Did you use R studio as well?

16)  Could you please explain how conditional inference trees differs from decision trees?

17)  Could you please summarize the data of lines 90-99 in a table?

18)  Could you please add the information of lines 101-115 in a table?

19)  I am sorry but I cannot find supplementary table 2.

20)  Data of lines 127-155 is very hard to read. Could you please add it in a table?

21)  In the machine learning decision tree analysis, were all the genes of the panel tested?  

22)  In Figure 1. Were all genes of the panel tested?

Reviewer 2 Report

The objective of this manuscript is to discover innovative prognostic biomarkers for pleural mesothelioma (PM), particularly those associated with cancer-associated fibroblasts (CAF) and desmoplastic stromal reaction (DSR). This is achieved through the utilization of H&E staining, IHC staining for FAP, and digital gene expression analysis of 76 genes. The study examines the correlation between DSR, FAP protein, selected gene expression, and the survival time of 77 PM patients. To predict the most effective gene candidates for distinguishing patient survival time, a Conditional Inference Tree (CIT) model is employed. While the work is intriguing, there are certain concerns regarding the experimental design and approach. Firstly, FAP alone is not an exclusive marker for CAF or DSR, as it can also be expressed in normal fibroblasts and PM cells due to the mesenchymal origin of PM cells. Merely adjusting the abundance through control samples is insufficient for distinguishing CAF from normal fibroblasts and PM cells. Secondly, there is a lack of an independent test set and functional investigation of the top gene candidates identified by CIT. The chosen cutoff for each gene in the survival analysis appears arbitrary and may not yield optimal results when applied to other datasets. Additionally, the reference to "unsupervised machine learning" in the title is unclear since CIT is a supervised model. Thirdly, the majority of genes in the CodeSet are not specific to a particular cell type. It remains uncertain which genes originate from CAFs, PM cells, or other cells, making the interpretation of results challenging. The study also overlooks the heterogeneity of CAFs since the digital gene expression analysis is conducted on bulk tumor tissue.

Round 2

Reviewer 1 Report

thank you for the answers

Reviewer 2 Report

Most of the concerns are solved in the revision